# Diagnosing Middle Ear Malformation by Pure-Tone Audiometry Using a Three-Dimensional Finite Element Model: A Case-Control Study

**DOI:** 10.3390/jcm12237493

**Published:** 2023-12-04

**Authors:** Shin-ichiro Kita, Toru Miwa, Rie Kanai, Yoji Morita, Sinyoung Lee, Takuji Koike, Shin-ichi Kanemaru

**Affiliations:** 1Department of Otolaryngology-Head and Neck Surgery, Kitano Hospital, Tazuke Kofukai Medical Research Institute, Osaka 5308480, Japan; s_kita@ent.kuhp.kyoto-u.ac.jp (S.-i.K.); r-kanai@kitano-hp.or.jp (R.K.); kanemaru@ent.kuhp.kyoto-u.ac.jp (S.-i.K.); 2Department of Otolaryngology-Head and Neck Surgery, Kyoto University, Kyoto 6068507, Japan; 3Department of Otolaryngology, Osaka Metropolitan University, Osaka 5458585, Japan; 4Department of Mechanical and Intelligent Systems Engineering, Graduate School of Informatics and Engineering, The University of Electro-Communications, Tokyo 1828585, Japant.koike@uec.ac.jp (T.K.); 5Department of Mechanical Engineering, Faculty of Engineering, Graduate Faculty of Interdisciplinary Research, University of Yamanashi, Yamanashi 4008510, Japan; leesinyoung@yamanashi.ac.jp

**Keywords:** tympanoplasty, ossicular chain, compliance, diagnostic criteria, finite element model

## Abstract

Background: Hearing loss caused by middle ear malformations is treated by tympanoplasty to reconstruct the acoustic conduction system. The mobility of the ossicles plays a crucial role in postoperative success. However, identifying the location of ossicular malformation based solely on preoperative audiograms is challenging due to the complex relationship between fixation location, deformity levels, and ossicular mobility. Methods: Middle ear finite element models for simulating ossicular malformations were created, and the results were compared with the actual preoperative audiograms. Results: This approach objectively diagnosed ossicular fixation and disarticulation, bypassing traditional criteria reliant on physician examination or imaging. Conclusion: This study suggests that future research should focus on developing a diagnostic framework utilizing large-scale data.

## 1. Introduction

The mammalian middle ear comprises three bones, namely the malleus, incus, and stapes, which are held in place by ligaments and tendons. This intricate structure serves as a conduit for transmitting sound vibrations between the tympanic membrane and the oval window. In individuals with an intact tympanic membrane and a well-aerated middle ear space, conductive hearing loss, characterized by a substantial air–bone (AB) gap in the audiometry test, is primarily attributed to the abnormalities in the ossicular chain, including fixation or discontinuity, as well as the presence of a third window within the inner ear [1].

The conditions that give rise to irregularities in the ossicular chain include congenital malformations, otosclerosis, trauma, and inflammatory diseases. As part of a typical development, the three ossicles naturally separate from the temporal bone and become interconnected [2,3,4]. The malformations commonly arise from factors such as the loss of the long process of the incus, sclerosis of the footplate of the stapes, or fixation of the malleus–incus complex to the attic wall of the middle ear [4]. Otosclerosis is characterized by abnormal bone remodeling around the stapes, leading to progressive stiffening of the annular ligament of the footplate. Traumatic injuries may result in the dislocation and separation of the incudomalleolar or incudostapedial joints between the ossicles, as well as ossicular fractures, which most frequently occur in the long process of the incus [5]. Given that over 80% of cases of conductive hearing loss can be attributed to conditions such as otitis media with cholesteatoma or chronic suppurative otitis media, the timely diagnosis of a congenital malformation or otosclerosis is often impeded. Similarly, delays in identifying traumatic injuries occur due to the high incidence of indirect injury [5,6]. Recognizing and comprehending the pathogenesis of ossicular chain defects are of paramount importance, as these defects can be rectified through surgical interventions.

Hearing loss due to ossicular malformations can be corrected with middle ear surgery. The intraoperative findings can influence the choice of surgical approach for ossicular chain reconstruction [7,8]. Computed tomography (CT), audiometry, and otoscopy are all used in preoperative studies. The simplest test to evaluate the health of the middle ear is otoscopy, which may identify middle ear fluid. The middle ear exerts a negative pressure when fluid is present, which increases the movement of the tympanic membrane. Since only the handle of the malleus is visible during otoscopy, it only offers a limited amount of information about the vibratory and continuity characteristics of the ossicles. The ossicles may be seen more clearly with this approach because of the recent advancements in the resolution of CT scans. However, it remains challenging to identify ossicular fixation and the ossicular chain’s whole or partial discontinuity with CT. Pure-tone audiometry (PTA) and tympanometry are two audiological procedures that may measure the effectiveness of sound transmission and impedance in the middle ear, and the findings of these tests have been shown to indicate the presence of middle ear disease. To quantify the microscopic and qualitative changes in middle ear disease, PTA, in particular, offers frequency-specific information.

PTA data have been used extensively to infer the pathology of middle ear disease, which has contributed to clinical practice. For instance, an up-sloping audiogram caused by a higher low-frequency threshold is characteristic of otosclerosis, and individuals with ossicular chain dislocation see significant increases in threshold across the board [9]. However, clinical data have been used to infer correlations between audiometry results and underlying disorders, and the precise impact of every type of pathology on the efficiency of sound transmission has not yet been thoroughly described. As a result, preoperative PTA has limited ability to identify the specific pathology that is impacting the ossicular chain.

Taking into consideration the diagnostic restrictions mentioned earlier, several studies have investigated the middle ear conduction system using the temporal bone model [10,11,12]. Recent studies have conducted a numerical analysis of the application of ossicular fixation and disarticulation in a middle ear finite element (FE) model to measure ossicular mobility [13,14,15,16,17]. Hirabayashi et al. reported the clinical relevance of these measurements in assessing the actual hearing pattern [18]; however, no consensus was reached, owing to the substantial variability among individuals. Moreover, this method has not yet been applied in the clinical setting.

The goal of this research was to compare the findings of each of these kinds of malformations with the actual audiograms in six patients and to examine whether these malformations could be inferred from preoperative audiograms. This study is a replication of the study by Hirabayashi et al. [18]. Based on the outcomes of the research, we propose that the utilization of PTA in association with an FE model would enhance the preoperative identification of the pathologies that underly conductive hearing loss, and this can be applied in the clinical setting.

## 2. Materials and Methods

### 2.1. Ethics

This study was approved by Kitano Hospital (approval no: 2001005; date of approval: 14 January 2020) and the Institutional Review Board of Kumamoto University, Japan (approval no: 2647; date of approval: 10 January 2023). This study adhered to the guiding principles of the Declaration of Helsinki. All patients—or their parents, if they were under 20 years of age and wanted to participate in this study—gave their informed consent.

### 2.2. Participants

The inclusion criteria were: (1) the patients had middle ear malformation confirmed at the time of surgery; (2) there was an absence of confounding pathologies, such as tympanic membrane lesions, cholesteatoma, and retrocochlear disease; and (3) the available” operative and audiometry reports were eligible for the study. Six patients with ossicular malformations who were diagnosed with middle ear malformation and had undergone surgical treatment at the two hospitals, between April 2009 and January 2020, were included in the analysis (Table 1). The patients (four men and two women) were 9–56 (34.0 ± 19.4; mean ± standard deviation) years of age. PTA was carried out in each patient. The patients were divided into groups as per Funasaka’s classification [4]: (1) those with incus–stapes joint avulsion due to dysplasia of the incus long limb or stapes superstructure; (2) those with stapes fixation; and (3) those with malleus head or incus body adhesion to the surrounding bone wall. PTA, tympanometry, and temporal bone-targeted CT were performed, and a three-dimensional (3D) image of the ossicles was obtained using Fusion Viewer, a component of a 3D image analysis system (SYNAPSE VINCENT; FUJIFILM Medical Co., Ltd., Tokyo, Japan). The images were automatically registered in a 3D space through normalized mutual information. This technique’s practicality and resilience have been verified using anatomical MR brain images, which included the introduction of noise and distortion [19]. To avoid mis-registration, the images were formed through the subtraction of the registration result from the CT axial images, which had a field of view of 25 cm encompassing the bilateral temporal bone regions. In the resulting images, the temporal reductions in CT values were represented as negative values (depicted in black), while increases were indicated as positive values (depicted in white). The creation of each image took approximately 1 min [20].

### 2.3. Theoretical Analysis

Modifications were made to the model proposed by Koike et al. [6], which generated FE models of the damaged middle ear for theoretical research purposes. The FE method is a numerical analysis method to numerically obtain approximate solutions for differential equations that are difficult to solve analytically. The domain in which the equations are defined is divided into subdomains (elements), and the equations in each subdomain are approximated by a relatively simple and common interpolation function. The tympanic membrane, ossicles, anterior malleal ligament (AML), posterior incudal ligament (PIL), stapedial annular ligament (SAL), tensor tympani tendon, and the stapedial muscle tendon are all components of this model. In this study, the superior malleal ligament (SML), lateral malleal ligament (LML), posterior malleal ligament (PML), and superior incudal ligament (SIL) were” constructed to simulate various pathological variations in the middle ear (Figure 1).

The volume velocity at the stapes footplate (V_SF_) was determined after applying a pure tone to the tympanic membrane surface for the sick ear model. The tone had a sound pressure of 80 dB and a frequency of 125–8000 Hz. As the sound pressure generated in the cochlea can be expressed as the product of the cochlear impedance and V_SF_, the ratio of the V_SF_ of the normal ear model to the diseased ear model, obtained using the following equation, corresponds to hearing loss:$$Ratio\;of\;V_{SF}\;(dB)=20log\left(\frac{V_{SF}\;of\;normal\;ear}{V_{SF}\;of\;diseased\;ear}\right)$$

Using this value, the results of hearing tests and numerical analyses were compared.

## 3. Results

### 3.1. Patients

The demographic features of all patients are displayed in Table 1. The audiograms and 3D-CT constructed images are shown in Figure 2a–e.

### 3.2. Comparison of Simulated and Actual Audiograms

#### 3.2.1. Incus–Stapes Joint Disarticulation due to Dysplasia of the Long Bone or Stapes Superstructure

Patient 1: The malformation from the long process of the incus to the vicinity of the stapes head was substituted by cords, the shape of which is shown in Figure 3a and Appendix A. The cords’ Young’s modulus, which is the constant of the proportionality between strain and stress in the coaxial direction in the elastic range where Hooke’s law holds, was altered in three stages from 1/10^3^ to 1/10^5^ of the normal value of the incus model (1.2 × 10^10^ N/m^2^). The shape was changed to 1/4 (Ver. 1) and 1/16 (Ver. 2) of the cross-sectional area of the long leg of the incus (Figure 3b). With respect to the stapes, the SAL Young’s modulus rose to 10 times that of their normal value (6.5 × 10^4^ N/m^2^) as otosclerosis was suspected (Figure 3c). A reduction in the Young’s modulus caused a decrease in the sound transmission efficiency (Figure 3b). In addition, the sound transmission efficiency was lower in the thinner cords (Ver. 2) compared with the thicker cords (Ver. 1) (Figure 3b). When a simulation was performed in thinner cords, the sound transmission efficiency decreased and the frequencies decreased (from 100 to 1000 Hz, Figure 3c). Based on the peak frequency, we assumed that Ver. 2 was able to reproduce the preoperative hearing of Patient 1 better; however, owing to the discrepancy in the auditory image, there was a ten-fold increase in the SAL Young’s modulus to reproduce the auditory image (Figure 3c). The SAL Young’s modulus varied significantly in the Ver. 2 model with a 1/10^4^-fold increase in the chordate’s Young’s modulus; the SAL’s modulus caused hearing loss in the low-frequency range (Figure 3d). Based on this finding, a Young’s modulus approximately eight times the SAL Young’s modulus of the Ver. 2 model and 1/10^4^ times the Young’s modulus of the cord-like object was appropriate to reproduce the auditory image of Patient 1 (Figure 2a and Figure 3d).

Patient 2: An audiogram reproduction was obtained using the Ver. 1 model with a Young’s modulus that was 1/10^5^ times greater than normal (Figure 2b and Figure 3c). In Patient 2, hearing loss in the low-frequency range was caused by bone conduction rather than air conduction, which was estimated by audiograms. A threshold >1000 Hz could be modified by the thickness and stiffness of the cords because the chordae transmit low-frequency vibrations [13]. From this point of view, Ver. 1 (1/10^5^) would best reproduce the auditory image in Patient 2.

#### 3.2.2. Stapes Fixation

Patient 3: The anterior crus of the stapes was eliminated from the normal model. Additionally, we assumed that the posterior crus was replaced by a cord-like substance with viscoelastic properties (Figure 4a and Appendix A). The viscoelastic properties were expressed using the generalized Maxwell model, and the storage modulus obtained from the model was used as the Young’s modulus of the cord-like material at each frequency [21]. Increasing the Young’s modulus resulted in a significant decrease in hearing at all frequencies; however, adding viscosity enabled the hearing image to appear similar to the actual auditory image of Patient 3 (Figure 2c and Figure 4b).

Patient 4: The Young’s modulus varied, and the hearing image was closer to that of Patient 4.

#### 3.2.3. Adhesion of the Head of the Malleus or the Body of the Incus to the Surrounding Bone Wall

Patient 5: Singular and combined fixation cases were simulated by the increase of the Young’s modulus of each ligament from 10 times (mild fixation) to 1000 times (severe fixation) of the normal value (Figure 5a and Appendix A). The normal values for each ligament were as follows: AML, 2.1 × 10^7^ N/m^2^; PIL, 6.5 × 10^5^ N/m^2^; SIL, 4.9 × 10^6^ N/m^2^; and SML, 4.9 × 10^6^ N/m^2^.

The ossification of the ligament was also considered using the ossicle’s Young’s modulus (1.2 × 10^10^ N/m^2^). The simulation of AML, PIL, SIL, and SML adhesions alone showed that while the Young’s modulus increased by a factor of 100, only a slight decrease was observed in the low-frequency range (Figure 5b). However, when ossification was assumed, the PIL, SML, AML, and SIL decreased in that order (Figure 5c). In particular, PIL fixation decreased in the low- to mid-frequency range (Figure 5c). When multiple ligaments of the malleus and incus adhered simultaneously, the frequency of the lower range decreased as the number of adherences increased (Figure 5d). When the degree of ligament fixation was varied at four levels (10-, 100-, and 1000-fold higher Young’s modulus, and ossification), hearing decreased progressively in the stiffer ligaments, especially at lower frequencies (Figure 5d). As a deviation from the auditory image was observed in Patient 5, the SAL Young’s modulus also varied, and the high-frequency range decreased, giving an appearance that was similar to the actual auditory image (Figure 2e and Figure 5d).

Patient 6: The intraoperative findings showed adherence to the AML, LML, SML, PML, SIL, and PIL, but not to the SAL, as well as a discrepancy between the actual auditory image and the simulated image, especially in the high-frequency range (Figure 2f and Figure 5d).

## 4. Discussion

The preoperative diagnosis of otoacoustic malformation was challenging before imaging techniques such as CT and 3D construction were developed and experimental tympanoplasty and other treatments such as tympanoplasty were performed; however, the advances in diagnostic techniques have improved the accuracy of diagnosing otoacoustic malformations [22]. In this study, audiometric simulations with 3D construction and engineering mechanics were used to compare the preoperative audiograms to evaluate the ossicular chain malformation. According to the findings of our research, the audiogram that was averaged and collected from clinical cases was mostly compatible with the audiometric pattern that was predicted for each of the diseases that were defined by our FE methods. The findings of this research will contribute to the development of analytical techniques for PTA that will make it easier to preoperatively determine the pathology causing conductive hearing loss.

In our current and previous studies [18], the AB gap is a valuable indicator of hearing loss severity attributed to middle ear conditions, as it compares the results of the air-conduction and bone-conduction hearing tests. An important observation from our study is that distinct pathologies manifest as varied AB gap shapes, characterized by features such as sloping, peaks, and dips. This suggests that these AB gap features might serve as diagnostic cues for middle ear conditions.

Regarding the sloping shape audiograms, we must consider the resonant frequency of the middle ear to comprehend why specific pathologies lead to distinct sloping audiograms. Wideband tympanometry allows an estimation of this frequency, which typically ranges from 800 to 1200 Hz in normal individuals [23]. Clinical audiograms are calibrated to display a flat line under normal hearing conditions, ensuring easy identification of hearing loss. Consequently, when bone conduction thresholds are normal, the audiogram exhibits no slope. A sloping audiogram arises from changes in the acoustic impedance (due to pathology) that shift the resonant frequency to higher or lower levels. For instance, ossicular dehiscence shifts the resonant frequency to a lower frequency (approximately 750 Hz), while otosclerosis shifts it to a higher frequency (approximately 1400 Hz) [24]. Stiffness, impacting low frequencies, and mass, influencing high frequencies, are the primary factors influencing impedance and resonant frequency [25]. A seesaw analogy has been employed to illustrate the interplay between stiffness and mass in shaping the audiogram [26]. In addition to sloping, some simulated and patient-obtained audiograms in our study exhibited peaks and dips at specific frequencies. While the peaks and dips are generally of limited clinical interest, their presence and frequency can offer crucial insights into the resonant frequency. As calibration values vary for each frequency, a shift in the resonant frequency may result in a non-smooth, calibrated curve, displaying a peak or a dip at a specific frequency. Therefore, assessing the peaks, dips, and slopes can provide valuable information to differentiate between different middle ear pathologies. Below, we delve into potential pathologies underlying the various types of slopes observed in audiograms.

Since the current FE method was unable to model the complete discontinuity of the ossicular chain, the examination of the discontinuity of the incus–stapes joint was limited to its partial discontinuity. In Patient 2, the separation of the long leg of the stapes or the stapes with dysplasia of the stapes superstructure was simulated. The stapes was connected by a relatively stiff fibrous cord, which was reproduced by altering “the Young’s modulus and the thinness of the cord in the middle ear FE model. In Patient 1, the actual auditory image was different from that obtained by altering only the Young’s modulus and fiber thinness, such that the rise in the SAL Young’s modulus caused a decrease in the low-frequency amplitude ratio. This finding suggests that the viscosity of the fibers may have contributed to the high-frequency amplitude. More detailed simulations and matching are expected in the future to allow for a more accurate determination. According to the findings of a previous research article, the simulated audiogram had a down-sloping and an increasing threshold with increasing degrees of discontinuity. An analysis carried out with the FE method revealed that a downwardly sloped audiogram was most prevalent, while the stiffness of the incudostapedial joint decreased and the ossicle mass increased. According to previous reports, one of the signs of incomplete discontinuity is an audiogram that slopes downward [27]. Hirabayashi et al. [18] stated that the audiograms of individuals with an incudostapedial joint discontinuity displayed broad variance, requiring independent investigations of each illness category, such as traumatic discontinuity and middle ear deformity. On the other hand, an increase in the mass of the ossicles may also cause a downward trend, although, from a clinical perspective, this might be caused by the attachment of granulation tissue, the presence of an inflammatory effusion, or the adherence of the matrix of a cholesteatoma. The results of this study were also consistent with previous studies regarding down-slope audiograms and middle ear malformations without mass components in the cavity of the middle ear. In future studies, a more accurate prediction of the ossicular chain can be made preoperatively for an accurate diagnosis by further modifying the thickness and viscosity of the fibrous cord-like structures.

In the stapes fixation simulation in Patient 3, where the stapes had a single leg and fibrotic union, the threshold was higher than expected if the stapes had remained rigid. However, the addition of a viscous component to the stapes produced a hearing image that was very similar to the actual hearing image. In Patient 4, the SAL was ossified, and the anterior leg was missing. In the middle ear finite element model, the results were consistent with the actual hearing with the stapes intact and rigid, and with the addition of SAL fixation. This finding was consistent with the intraoperative findings. It was also similar to the experimental findings, wherein the velocity of the ossicles decreased significantly in response to the sound input when the SAL of the temporal bone specimen had been artificially immobileized. Therefore, more detailed parameter changes are necessary for an accurate diagnosis using auditory images. Finally, in the simulation of the case of Patient 5 (adhesion of the head of the malleus or the incus bone body to the surrounding bone wall), the simultaneous adhesion of multiple ligaments of the malleus and incus bones tended to progressively decrease as several adhesion points increased, especially at a low-frequency range. The stiffness of the fixation also progressively decreased as the ligament stiffness increased, mainly at lower frequencies; the SAL Young’s modulus also decreased at higher frequencies. These results were consistent with the intraoperative findings. Clinically, it is known that otosclerosis (fixation of the stapes footplate) results in an upward-sloping audiogram, and an upward-sloping audiogram would also be anticipated if the stiffness component arises as a result of the malleus and incus attic fixation [9]. Our findings were consistent with previous studies regarding up-slope audiograms and stapes/attic fixation.

It was difficult to simulate the case of Patient 6 Ie of discrepaIcies in the auditory images. These discrepancies were attributed to the inability to include the parameters for assessing external auditory canal malformations. Based on the above results, it is difficult to estimate the shape of the ossicular malformation from the auditory image alone owing to the high number of variations. However, if the shape, continuity, and attachment of the ossicles to the tympanic chamber are diagnosed based on the CT and 3D constructed images, a more accurate preoperative diagnosis is possible. Conversely, the use of a middle ear finite element model can help to predict the possible findings when performing a hearing test, and the use of a combination of images can assist in a more detailed diagnosis. This study will be useful for clinics that do not have CT equipment. The analysis of mobility in each patient using the FE model provides theoretical support for a diagnosis based on mobility because the results were obtained under identical conditions without individual differences or other diseases.

This study is limited by its relatively small sample size and low level of evidence. Therefore, it will be necessary to include more patients in future studies. In addition, it was not possible to assess for the external auditory canal malformations in this study. Qi et al. constructed and reported the FE model of the normal newborn ear canal [28]. However, the reliability of subjective audiometry is lower in pediatric patients. Thus, the FE model of the normal adult external ear canal can be constructed, and malformations of the adult external ear canal can be analyzed in the near future.

## 5. Conclusions

Using the middle ear FE model to quantify the mobility of the ossicles during detachment and ligament fixation, we evaluated the changes in six patients with middle ear malformations depending on the location and degree of fixation, and compared the results with their actual audiograms, proving that middle ear malformations can be inferred from the preoperative audiograms. Furthermore, we verified an objective method of diagnosing fixation and detachment of the ossicles, instead of using the conventional diagnostic criteria developed based on the findings of the examining physician or the CT and 3D-constructed images. Future studies should develop a diagnostic scheme using large-scale data to validate our findings and implement them in clinical practice.

## Figures and Tables

**Figure 1 jcm-12-07493-f001:**
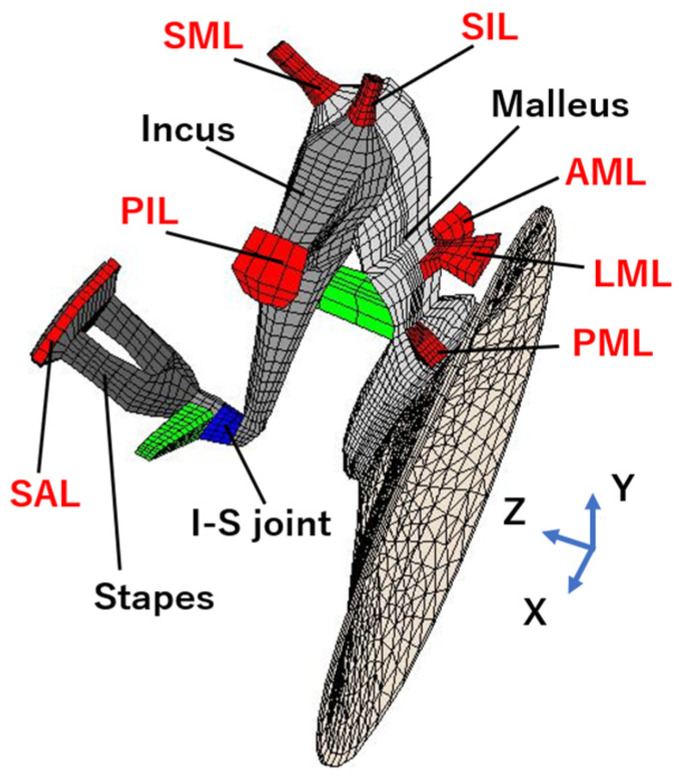
Finite element model of the human middle ear. AML: anterior malleal ligament; PIL: posterior incudal ligament; SIL: superior incudal ligament; SAL: stapedial annular ligament; LML: lateral malleal ligament; SML: superior malleal ligament; PML: posterior malleal ligament.

**Figure 2 jcm-12-07493-f002:**
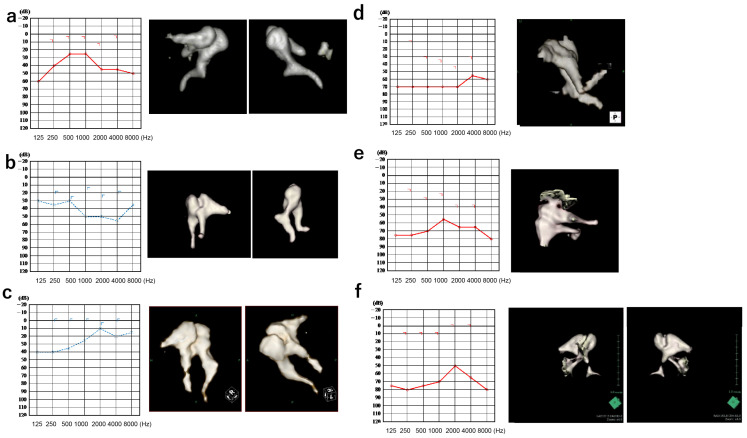
Audiograms and ossicular images of 3D-CT scans of every patient. (**a**) Patient 1, (**b**) Patient 2, (**c**) Patient 3, (**d**) Patient 4, (**e**) Patient 5, and (**f**) Patient 6. Red line indiceted right ear audiogram. Blue line indicated left ear audiogram.

**Figure 3 jcm-12-07493-f003:**
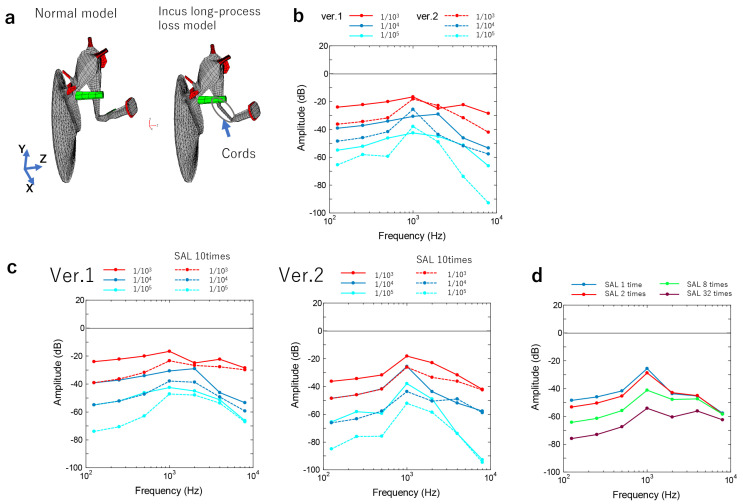
Simulation of the incus–stapes joint malformations. (**a**) FE model of the incus–stapes joint malformation. (**b**) Calculating audiograms by altering the cords’ Young’s modulus (1/10^3^–1/10^5^) and the shape of the cords’ cross-sectional area (Ver. 1: 1/4, Ver. 2: 1/16). (**c**) Calculating audiograms by changing the SAL Young’s modulus 10 times in Ver. 1 and Ver. 2. (**d**) Calculating audiograms by altering the SAL Young’s modulus with a Young’s modulus 1/10^4^ times greater than that of the cords.

**Figure 4 jcm-12-07493-f004:**
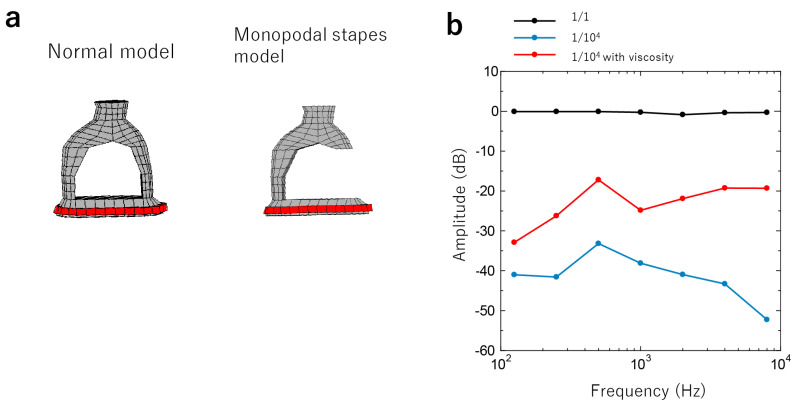
Simulation of the stapes malformation. (**a**) FE model of the malformation of stapes. (**b**) Calculating audiograms by altering the cord-like object’s Young’s modulus and adding viscosity.

**Figure 5 jcm-12-07493-f005:**
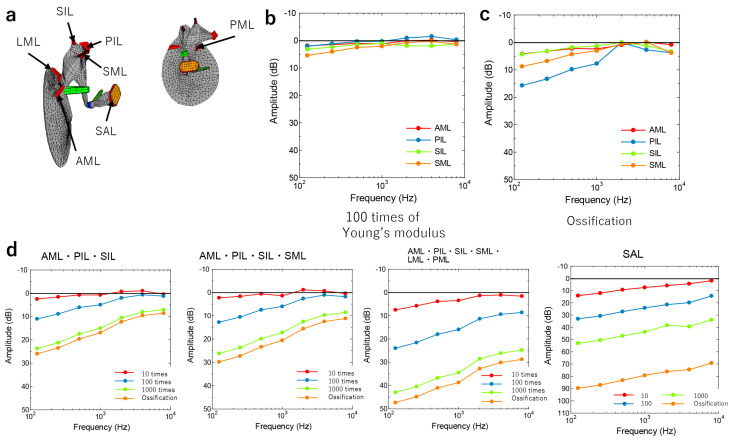
Simulation of the malleus and incus malformation. (**a**) FE model of the malformation around the malleus and incus. (**b**) Calculating audiograms by changing the “Young’s modulus of every ligament to 100 times the normal value. (**c**) Calculating audiograms by changing the Young’s modulus of each ligament to that at ossification. (**d**) Calculating audiograms by changing the Young’s modulus of each ligament to that at” combined fixation.

**Table 1 jcm-12-07493-t001:** Clinical characteristics of the included patients.

	Category	Age	Sex	PTA 4 (dB HL)	AB Gap (dB HL)	TG	SR	DPOAE	CT	Diagnosis	Surgery
1	Incus	24	M	31.7	27.5	Ad	+	-	None	Incus long process, complete loss	Tympanoplasty IIIiM
2	Incus	47	M	43.3	25.0	Ad	-	-	None	Incus long process, incomplete loss	Tympanoplasty III
3	Stapes	9	F	23.3	18.8	A	-	NA	None	Monopodal stapes	Tympanoplasty IVi
4	Stapes	50	M	70.0	30.0	A	-	NA	STD around stapes	Monopodal stapes Adhesion to the petrosus	TympanoplastyI
5	Adhesion	56	M	23.3	28.8	A	-	-	Adhesion of malleus to tympanic cavity	Adhesion of the malleus to the tympanic cavity Fixation of stapes	Tympanoplasty+ Stapedotomy
6	Adhesion + abnormal EAC	18	F	65.0	60.0	A	NA	NA	Narrow EAC Adhesion of malleus and incus	Narrow EAC Adhesion of malleus and incus	Tympanoplasty IIIc

Legend: PTA 4: Pure-tone audiometry 4; TG: Tympanogram; Ad: Type A discontinuity; A: Type A; SR: Stapedial reflex; DPOAE: Distortion product otoacoustic emissions; EAC: External ear canal; M: Male; F: Female; dB HL: decibel hearing level; AB gap: Air–bone gap; NA: Not applicable; CT: Computed tomography; STD: Soft tissue density; Tympanoplasty type III: reconstruction of ossicular chain; Type IIIiM: subtype of type III; Type IIIc: subtype of type III; Type IVi: subtype of tympanoplasty.

## Data Availability

Data can be made available from the corresponding author upon reasonable request.

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
