# Peer review of "Diagnosing Middle Ear Malformation by Pure-Tone Audiometry Using a Three-Dimensional Finite Element Model: A Case-Control Study"

_jcm, 2023, doi:10.3390/jcm12237493_

Round 1
Reviewer 1 Report
Comments and Suggestions for Authors
I have had the privilege of reviewing the manuscript titled “Diagnosing middle ear malformation by pure-tone audiometry using a computational sound transmission model: A case-control study”. The study has an interesting future area using simulation of ossicular malformations by creating a middle ear model.
Overall, I find the study to be well-conceived.
The cohort size is small with 6 patients, but on the other hand sufficient for a case-control study. The study's objectives are clear. The study has fine and clear images in the figures, which help the reader understand the results better.
In line 107 the authors write Figure 1A-F, but I think it is correct to write Figure 2?
In line 112, I have a question about the retrospective nature of the study in which patients underwent surgical treatment between 2009 and 2020, if 3D image was used as a method already then?
In line 118-119 “…a field of view (FOV)” is unnecessary because the manuscript does not use this more than ones.
In line 124, Table 1 – there is a need of more text in caption for example all abbreviations such as ABgap, TG, SR. Regarding Threshold (dB) in Table 1, is it PTA4? Add more information about these in caption.
The manuscript has many abbreviations, maybe use a list?
In line 150 you write 2a-e – but in caption line 152-153 is used (i-vi) – use consistent explanations in the whole manuscript.
In line 159 “The cords Young’s modulus” maybe need an explanation somewhere in manuscript.
In line 174 “Based on…, we believe…” – is this Results or maybe move to Discussion?
Same in line 183 “From this point of view, we assumed…” – in Results only results and in Discussion what you believe or assume.
In line 186, the abbreviation FE-model need some explanation, and IS.
In line 246, word “FEM”, need explanation, can’t find it in manuscript.
From a language perspective, the manuscript is excellently written and requires no further editing.
Author Response
I have had the privilege of reviewing the manuscript titled “Diagnosing middle ear malformation by pure-tone audiometry using a computational sound transmission model: A case-control study”. The study has an interesting future area using simulation of ossicular malformations by creating a middle ear model.
Overall, I find the study to be well-conceived.
The cohort size is small with 6 patients, but on the other hand sufficient for a case-control study. The study's objectives are clear. The study has fine and clear images in the figures, which help the reader understand the results better.
In line 107 the authors write Figure 1A-F, but I think it is correct to write Figure 2?
>We thank the reviewer for this comment. The reviewer is correct in pointing out that line 107 should have cited Figure 2. However, this description has been removed in the revised manuscript as it was deemed unnecessary.
In line 112, I have a question about the retrospective nature of the study in which patients underwent surgical treatment between 2009 and 2020, if 3D image was used as a method already then?
> We thank the reviewer for this comment. We can confirm that 3D images were used before the surgery.
In line 118-119 “…a field of view (FOV)” is unnecessary because the manuscript does not use this more than ones.
> We thank the reviewer for this comment. We have made the relevant correction accordingly.
In line 124, Table 1 – there is a need of more text in caption for example all abbreviations such as ABgap, TG, SR. Regarding Threshold (dB) in Table 1, is it PTA4? Add more information about these in caption.
> We thank the reviewer for this comment. We have added the relevant caption to the revised manuscript accordingly.
The manuscript has many abbreviations, maybe use a list?
> We thank the reviewer for this comment. We have added a list of abbreviations in the manuscript.
In line 150 you write 2a-e – but in caption line 152-153 is used (i-vi) – use consistent explanations in the whole manuscript.
> We thank the reviewer for this comment. We have corrected the relevant content accordingly.
In line 159 “The cords Young’s modulus” maybe need an explanation somewhere in manuscript.
> We thank the reviewer for this comment. We have added an explanation on Young’s modulus, which is the constant of proportionality between strain and stress in the coaxial direction in the elastic range where Hooke's law holds.
In line 174 “Based on…, we believe…” – is this Results or maybe move to Discussion?
> We thank the reviewer for this comment. This sentence was placed in the Results section because it describes the verification of the consistency of our simulations. We have reworded this description as following in response to this comment:
“Based on this finding, a Young's modulus approximately eight times the SAL modulus of the Ver. 2 model and 1/104 times the Young's modulus of the cord-like object was appropriate to reproduce the auditory image of Patient 1 (Figures 2a and 3d).”
Same in line 183 “From this point of view, we assumed…” – in Results only results and in Discussion what you believe or assume.
> We thank the reviewer for this comment. We have reworded this description as following:
“From this point of view, Ver. 1 (1/105) would best reproduce the auditory image in Patient 2.”
In line 186, the abbreviation FE-model need some explanation, and IS.
> We thank the reviewer for this comment. “IS” refers to incus-stapes. We have corrected “IS” to “incus-stapes” accordingly. The “FE-model” refers to finite-element models. We have added an explanation accordingly as follows:
“The FE method is a numerical analysis method to numerically obtain approximate solutions for differential equations that are difficult to solve analytically. The domain in which the equations are defined is divided into subdomains (elements), and the equations in each subdomain are approximated by a relatively simple and common interpolation function.”
In line 246, word “FEM”, need explanation, can’t find it in manuscript.
> We thank the reviewer for this comment. “FEM” refers to finite-element methods. This is equivalent to the FE model.
From a language perspective, the manuscript is excellently written and requires no further editing.
Reviewer 2 Report
Comments and Suggestions for Authors
The article is well-written; however, there are important details that must be reviewed. Therefore, I would like to indicate some points that could be improved. See below:
1) The term "computational sound transmission model" was used only in the title. Therefore, I recommend that the title be rethought. See that the term used throughout the paper was finite element model. This way, you need to use this terminology (three-dimensional finite element mode, - FEM).
2) In Table 1, I recommend inserting the legend about this itens (ABgap, TG, SR, DPOAE, CT, M, F, Ad, A, NA, STD, EAC, ⅢiM, Ⅲ, Ⅳi, I, IIIc). The reader needs to understand all the information in the table without needing to return to the text.
3) Review the abbreviation FEM in the text. Before abbreviating, you must make a correlation with the word.
4) On page 264, review the terminology "Wideband absorptiometry (WBA)". Nowadays, we have these terminologies updated, such as Wideband tympanometry, and wideband absorbance. So rearrange this information in the text.
5) Review the abbreviation IS in the text. Before abbreviating, you must make a correlation with the word.
6) In conclusion, it´s very important to reorganize the information. Remember what the goal of your research was? (The goal of the current research was to compare the findings of each of these kinds of malformations with the actual audiograms in six patients and to examine whether these malformations could be inferred from preoperative audiograms) So, you need to start answering your initial goal. The writing of your conclusion is not adequate for your initial goal. Please review this point.This way, either you change your goal or you change your conclusion. See, it is very important to be reviewed.
Author Response
The article is well-written; however, there are important details that must be reviewed. Therefore, I would like to indicate some points that could be improved. See below:
1) The term "computational sound transmission model" was used only in the title. Therefore, I recommend that the title be rethought. See that the term used throughout the paper was finite element model. This way, you need to use this terminology (three-dimensional finite element mode, - FEM).
> We thank the reviewer for this comment. We changed the title accordingly and included the term “three-dimensional finite element model.”
2) In Table 1, I recommend inserting the legend about this itens (ABgap, TG, SR, DPOAE, CT, M, F, Ad, A, NA, STD, EAC, ⅢiM, Ⅲ, Ⅳi, I, IIIc). The reader needs to understand all the information in the table without needing to return to the text.
> We thank the reviewer for this comment. We have added the relevant legend to the revised manuscript describing the tests and surgery.
3) Review the abbreviation FEM in the text. Before abbreviating, you must make a correlation with the word.
> We thank the reviewer for this comment. We have corrected the relevant content accordingly and reworded “FEM” to “finite-element (FE)” methods.
4) On page 264, review the terminology "Wideband absorptiometry (WBA)". Nowadays, we have these terminologies updated, such as Wideband tympanometry, and wideband absorbance. So rearrange this information in the text.
> We thank the reviewer for this comment. “Wideband absorptiometry” has been changed to “Wideband tympanometry” accordingly.
5) Review the abbreviation IS in the text. Before abbreviating, you must make a correlation with the word.
> We thank the reviewer for this comment. We have corrected the relevant content accordingly and reworded “IS” to “incus-stapes.”
6) In conclusion, it´s very important to reorganize the information. Remember what the goal of your research was? (The goal of the current research was to compare the findings of each of these kinds of malformations with the actual audiograms in six patients and to examine whether these malformations could be inferred from preoperative audiograms) So, you need to start answering your initial goal. The writing of your conclusion is not adequate for your initial goal. Please review this point.This way, either you change your goal or you change your conclusion. See, it is very important to be reviewed.
> We thank the reviewer for this comment. The conclusions section has been revised as follows:
“Using the middle ear FE model to quantify the mobility of the ossicles during detachment and ligament fixation, we evaluated the changes in six patients with middle ear malformations depending on the location and degree of fixation, and compared the results with their actual audiograms, proving that middle ear malformations can be inferred from the preoperative audiograms.”
Round 2
Reviewer 2 Report
Comments and Suggestions for Authors
Dear authors,
I was very pleased with the corrections made.
At this time, I recommend minor adjustments to the caption for table 1.
First, below table 1 write the word legend and rearrange in this way:
Legend: PTA 4: Pure-tone audiometry 4); TG: Tympanograms; Ad: Type A discontinuity; A: Type A; SR: Stapedial reflex; DPOAE: Distortion product otoacoustic emissions; EAC: External ear canal; M: Male; F: Female; dB HL: decibel hearing level; AB gap: Air-bone gap; NA: Not applicable; CT: Computed tomography; STD: Soft tissue density; Tympanoplasty type III: reconstruction of ossicular chain ; Type IIIiM: subtype of type III; Type IIIc: subtype of type III; Type IVi: subtype of tympanoplasty.
Please check that all items have been properly mentioned.
Author Response
Respond to reviewer
Reviewer 2
Dear authors,
I was very pleased with the corrections made.
At this time, I recommend minor adjustments to the caption for table 1.
First, below table 1 write the word legend and rearrange in this way:
Legend: PTA 4: Pure-tone audiometry 4); TG: Tympanograms; Ad: Type A discontinuity; A: Type A; SR: Stapedial reflex; DPOAE: Distortion product otoacoustic emissions; EAC: External ear canal; M: Male; F: Female; dB HL: decibel hearing level; AB gap: Air-bone gap; NA: Not applicable; CT: Computed tomography; STD: Soft tissue density; Tympanoplasty type III: reconstruction of ossicular chain ; Type IIIiM: subtype of type III; Type IIIc: subtype of type III; Type IVi: subtype of tympanoplasty.
Please check that all items have been properly mentioned.
> Thanks for your comments. I revised them as you mentioned.